# Metformin regulates bone marrow stromal cells to accelerate bone healing in diabetic mice

Yuqi Guo[1], Jianlu Wei[2], Chuanju Liu[2], Xin Li[1,3,4]*, Wenbo Yan[1]*

[1]Department of Molecular Pathobiology, New York University College of Dentistry, New York, United States; [2]Department of Orthopedic Surgery, NYU Grossman School of Medicine, New York, United States; [3]Laura and Isaac Perlmutter Cancer Center, NYU Grossman School of Medicine, New York, United States; [4]Department of Urology, NYU Grossman School of Medicine, New York, United States

**Abstract** Diabetes mellitus is a group of chronic diseases characterized by high blood glucose levels. Diabetic patients have a higher risk of sustaining osteoporotic fractures than non-diabetic people. The fracture healing is usually impaired in diabetics, and our understanding of the detrimental effects of hyperglycemia on fracture healing is still inadequate. Metformin is the first-line medicine for type 2 diabetes (T2D). However, its effects on bone in T2D patients remain to be studied. To assess the impacts of metformin on fracture healing, we compared the healing process of closed-wound fixed fracture, non-fixed radial fracture, and femoral drill-hole injury models in the T2D mice with and without metformin treatment. Our results demonstrated that metformin rescued the delayed bone healing and remolding in the T2D mice in all injury models. In vitro analysis indicated that compromised proliferation, osteogenesis, chondrogenesis of the bone marrow stromal cells (BMSCs) derived from the T2D mice were rescued by metformin treatment when compared to WT controls. Furthermore, metformin could effectively rescue the impaired detrimental lineage commitment of BMSCs isolated from the T2D mice in vivo as assessed by subcutaneous ossicle formation of the BMSC implants in recipient T2D mice. Moreover, the Safranin O staining of cartilage formation in the endochondral ossification under hyperglycemic condition significantly increased at day 14 post-fracture in the T2D mice receiving metformin treatment. The chondrocyte transcript factors SOX9 and PGC1$\alpha$, important to maintain chondrocyte homeostasis, were both significantly upregulated in callus tissue isolated at the fracture site of metformin-treated MKR mice on day 12 post-fracture. Metformin also rescued the chondrocyte disc formation of BMSCs isolated from the T2D mice. Taken together, our study demonstrated that metformin facilitated bone healing, more specifically bone formation and chondrogenesis in T2D mouse models.

*For correspondence:
xl15@nyu.edu (XL);
wy12@nyu.edu (WY)

Competing interest: The authors declare that no competing interests exist.

## Editor's evaluation

This fundamental work advances our understanding of the effects of metformin on bone healing in hyperglycemic conditions. The evidence supporting the conclusion is convincing, using three different types of bone fracture models in type-2 diabetes (T2D) mice. This paper is of potential interest to skeletal biologists, orthopaedic surgeons, and endocrinologists who study the effects of metformin on fracture healing.

## Introduction

Diabetes mellitus is a group of chronic diseases characterized by high blood glucose levels. It is estimated that more than 347 million people worldwide currently have diabetes and many more people are estimated to become diabetic soon (*Danaei et al., 2011*). Among the diabetic patients, more than 90% are suffering from type 2 diabetes (T2D), which is caused by insulin resistance in peripheral tissues. Many tissues, including the skeleton, will be adversely affected by hyperglycemia if not controlled (*Yan and Li, 2013*). Both type 1 diabetes and T2D are associated with an increased risk of osteoporosis and fragility fractures (*Yan and Li, 2013*). It is recognized that oral antidiabetic medicines affect bone metabolism and turnover (*Yan and Li, 2013*). As an insulin sensitizer, patients with T2D are frequently prescribed metformin. In T2D patients, metformin treatment was associated with a decreased risk of bone fracture (*Vestergaard et al., 2005*). The osteogenic effects of metformin have been documented in both cellular and rodent models. Metformin promoted osteoblast differentiation and inhibited adipocyte differentiation in rat bone marrow mesenchymal stem cells culture (*Gao et al., 2008*). In rat primary osteoblasts culture, metformin increased trabecular bone nodule formation (*Shah et al., 2010*). In ovariectomized mice, metformin was also shown to improve the compromised bone mass and quality (*Gao et al., 2010*). Furthermore, in streptozotocin-induced type 1 DM model, metformin stimulated bone lesion regeneration in rats (*Molinuevo et al., 2010*). However, to date, there is no study on the skeletal effects of metformin in a T2D model. A recent study found that the incidence of total knee replacement over 4 y was 19% lower among patients with type 2 diabetes who were regular metformin users compared with non-users (*Lai et al., 2022*). In addition to reducing glucose levels, metformin may modulate inflammatory and metabolic factors, leading to reduced inflammation and plasma lipid levels (*Lai et al., 2022*). Better knowledge of how metformin treatment influences skeletal tissues under T2D condition is of great clinical relevance in view of the fast-growing population of patients with T2D.

MKR mouse model was generated and characterized by LeRoith and colleagues and expresses a dominant negative mutant of human IGFI receptor specifically in skeletal muscles (*Fernández et al., 2001*). The expression of this dominant negative mutant human IGFI receptor decreased glucose uptake and causes insulin resistance, and the MKR transgenic mouse rapidly develops severe diabetes (*Fernández et al., 2001*). This mouse model has been widely used in T2D research. In this study, the effects of metformin on bone cell lineage determination and regeneration were characterized using the MKR mouse model.

## Results

### Metformin promotes healing in fracture models under hyperglycemic conditions

In order to evaluate the fracture healing that involves endochondral ossification in mice, we adapted this well-accepted Bonnarens and Einhorn fracture mouse model (*Figure 1A*). Animals were sacrificed on day 14, 23, or 31 post-fracture representing the inflammatory stage, the endochondral stage, and the remodeling stage during the femur fracture repair process (*Einhorn and Gerstenfeld, 2015*). As shown in *Figure 1—figure supplement 1*, 12-week-old WT and MKR mice were treated with PBS or metformin through daily intraperitoneal injection. Blood glucose level (*Figure 1—figure supplement 1A*) and fasting glucose tolerant test (*Figure 1—figure supplement 1B*) showed that metformin significantly reduced glucose levels in MKR mice as anticipated. Meanwhile, metformin-treated MKR mice exhibited similar healing and remodeling speed to that of the WT groups throughout the entire healing process, and the fractured bones were completely healed on day 31 (*Figure 1B*). On the other hand, the healing process in the MKR-PBS group was much delayed, with a visible amount of callus tissue remained at the fracture site on day 31. Additionally, histological analysis of the callus area (*Figure 1C*) disclosed a postponed peak callus formation in the MKR-PBS group when compared to WT-PBS, WT-met, and MKR-met groups on day 23 post-fracture, as well as a delayed remodeling on day 31 post-fracture. A quantitative analysis of the callus at the fracture area (*Figure 1D*) confirmed that metformin significantly enhanced bone healing in the T2D mice compared to the PBS-treated MKR mice. We also harvested the callus tissue at fracture sites on days 12 and 21 post-fracture to examine the expression of genes that contribute to adipogenesis, osteogenesis, and chondrogenesis (*Figure 1—figure supplement 2*). A significant reduction in adipogenesis transcription factor

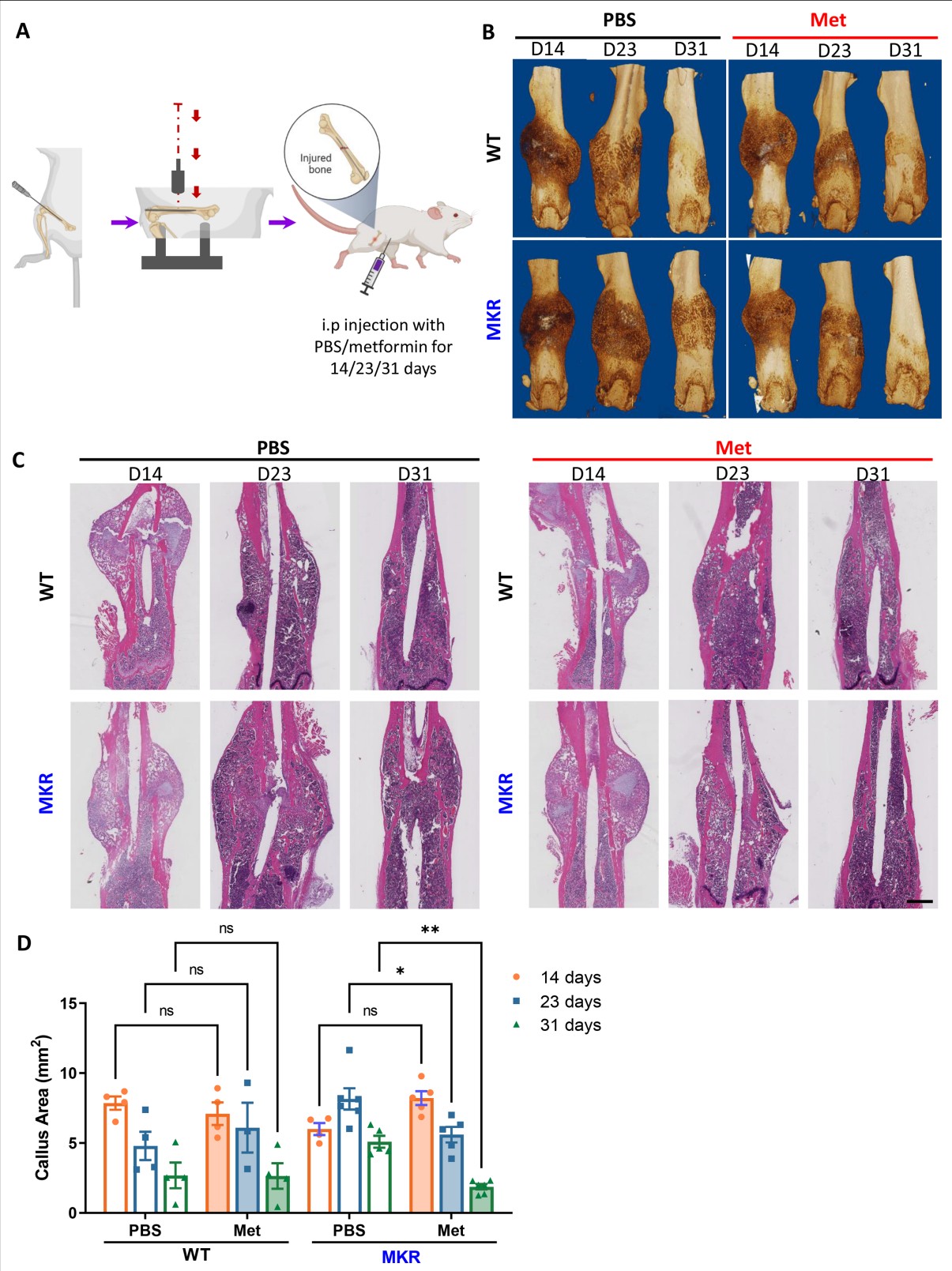

**Figure 1.** Improved healing in closed transverse fracture. (**A**) Schematic representation of the experimental design (created with BioRender. com). (**B**) Representative µCT images of femurs from each treatment group. (**C**) H&E staining of longitudinal femur sections (scale bar, 1 mm). (**D**) Histomorphometry analysis was performed on those H&E slides to evaluate the callus area at the fracture site from each treatment group (ANOVA, followed by Tukey's post hoc test), Bars show mean ± SEM. N = 3–6, *p<0.05, **p<0.005.

*Figure 1 continued on next page*

*Figure 1 continued*

The online version of this article includes the following source data and figure supplement(s) for figure 1:

**Source data 1.** Source data for *Figure 1D*.

**Figure supplement 1.** Metformin's effect on glucose levels in MKR mice.

**Figure supplement 1—source data 1.** Source data for *Figure 1—figure supplement 1A and B*.

**Figure supplement 2.** Transcript factor expressions within the femoral fracture callus tissue.

**Figure supplement 2—source data 1.** Source data for *Figure 1—figure supplement 2A–H*.

accompanied with a significant induction of chondrogenesis transcript factor SOX9 was detected in the callus harvested on day 12 post-fracture. SOX9 is a master transcription factor that plays a key role in chondrogenesis (*Kolettas et al., 2001*). In addition, at day 21 post-fracture, PGC1α, which is required for chondrocyte metabolism and cartilage homeostasis (*Kawakami et al., 2005*; *Zhao et al., 2014*), was significantly upregulated in the metformin-treated MKR mice.

A similar effect of metformin was also observed in a non-fixed radial fracture model (*Figure 2*). After non-fixed radial fracture was introduced, the animal was treated with either PBS or metformin for 14 or 23 d (*Figure 2A*). In the WT groups at 14 d post-fracture, both PBS and metformin-treated animals started to exhibit sufficient amount of callus with the sign of bridging of the fracture ends (*Figure 2B*). In the 14-day post-fracture MKR mice, those treated with metformin exhibited more healing callus than the PBS-treated ones with a significantly greater percentage of callus bridging at the fracture site (*Figure 2C*). Bone mineral density was also higher in the metformin-treated MKR mice when compared to the PBS-treated MKR mice (*Figure 2D*). Bone volume/tissue volume ratio (*Figure 2E*) and trabecular thickness (*Figure 2F*) were not regulated by metformin in either WT or MKR mice. In the 23-day post-fracture groups, only PBS-treated MKR animals failed to show callus formation and bridging at the fracture site compared to all the other groups (*Figure 2G and H*), indicating a delayed healing process in the MKR PBS group. Bone mineral density levels in the metformin-treated MKR mice were similar to the PBS-treated WT mice and were significantly higher than the PBS-treated MKR mice (*Figure 2I*). In the 23-day post-fracture groups, the bone volume/tissue volume ratio (*Figure 2J*) and trabecular thickness (*Figure 2K*) were significantly improved by metformin in MKR mice compared to the PBS-treated MKR mice. These data suggest that 23 d of metformin treatment favorably affected bone healing. The beneficial effects of 23 d of metformin treatment on the callus bridging (*Figure 2H*), BV/TV% (*Figure 2J*), and Tb. Thickness (*Figure 2K*) became significant in contrast to the trends of improvement observed in the mice received short-term treatment for 14 d (*Figure 2C, E and F*).

## Metformin improves healing in the drill hole injury model under hyperglycemic conditions

To further investigate metformin's effect on bone repair, a drill-hole model was established (*Figure 3A*). After 14-day treatment with PBS or metformin, we examined the injury site using μCT scanning. Exterior and interior general views of the femur indicate a non-filled open wound exhibited only in the PBS-treated MKR group, while all the other three groups showed mainly filled holes indicating a faster healing speed (*Figure 3B–D*). Tracing of the drill-hole injury sites presented a detailed view of the callus tissues formed within the drill hole (*Figure 3D*). In the WT animals, quantitative analysis of the μCT images showed no difference between the PBS and the metformin treatments. Significant lower BMD (*Figure 3E*) and BV/TV ratio (*Figure 3F*) were observed in PBS-treated MKR mice when compared to the metformin-treated ones. The PBS-treated MKR mice also demonstrated prominently higher bone porosity (*Figure 3G*) and total pore space (*Figure 3H*) within the callus tissue, suggesting that delayed bone healing and remodeling in MKR mice was rescued by metformin.

## Metformin accelerates bone formation under hyperglycemic conditions

In order to examine metformin's effects on bone formation in vivo, we conducted bone Alizarin Red and Calcein Double Labeling injections on WT and MKR mice. *Figure 4A* suggests that the florescent labels in WT mice remained the same between the PBS- and metformin-treated animals. In contrast, the distance between the two labels was greater in the metformin-treated than the PBS-treated MKR mice. This observation was supported by serum levels of amino-terminal propeptide of type

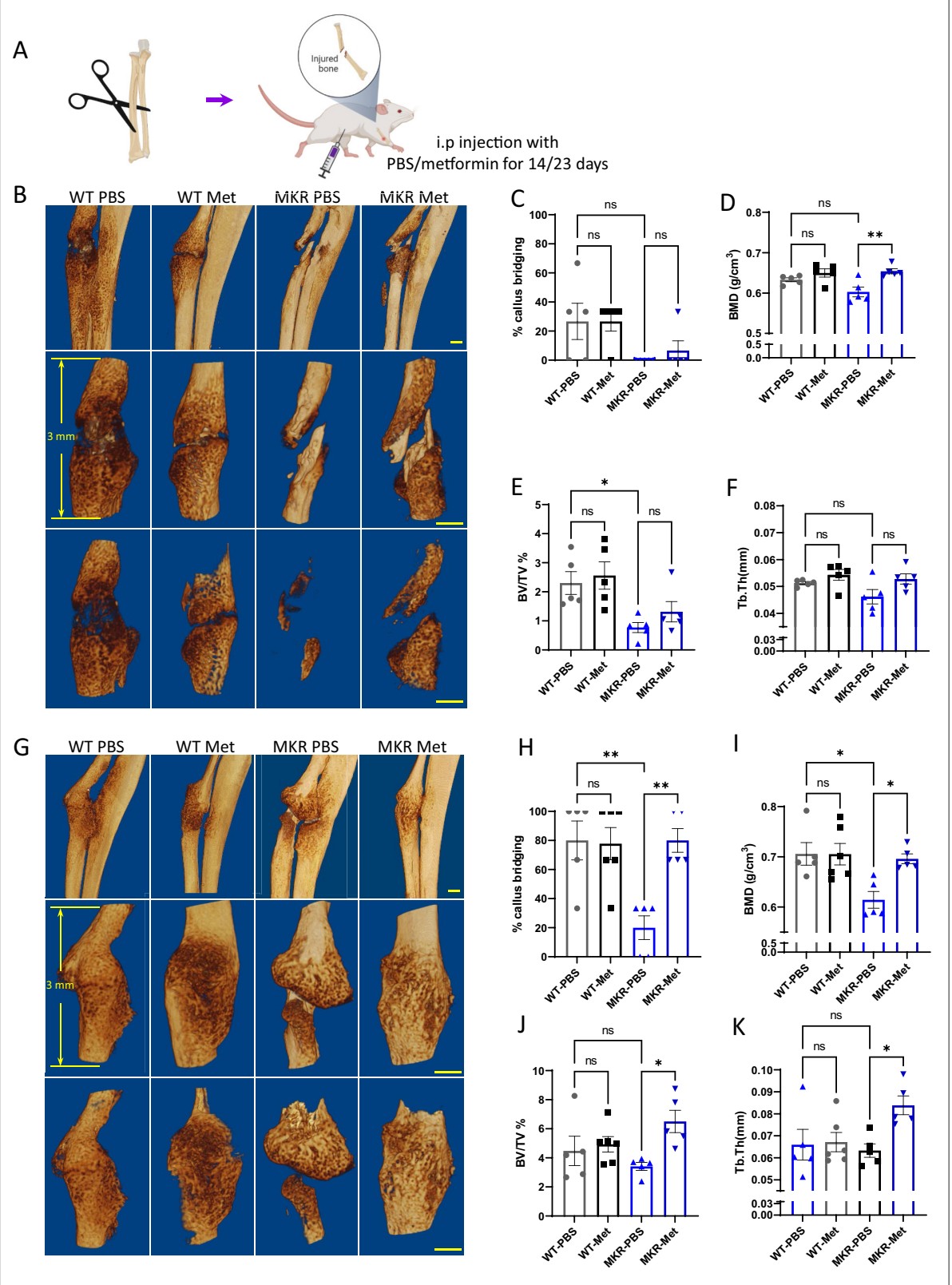

**Figure 2.** Improved healing in non-fixed radial fracture at two different time points (14 d and 23 d post-fracture). (**A**) Schematic representation of the experimental design (created with BioRender.com). (**B**) Representative µCT images of mouse radiuses (top) and fracture sites (bottom) 14 d post-fracture (scale bar, 500 µm). Measured parameters (**C–F**) by µCT at 14 d post-fracture. (**C**) Percentage of callus bridging (%). (**D**) Bone mineral density (BMD; g/cm³). (**E**) Bone volume/tissue volume (BV/TV; %). (**F**) Trabecular thickness (Tb.Th; mm). (**G**) Representative µCT images of mouse radiuses (top) and

*Figure 2 continued on next page*

Figure 2 continued

fracture sites (bottom) 23 d post-fracture (scale bar, 500 µm). Measured parameters (**H–K**) by µCT at 23 d post-fracture. (**H**) Percentage of callus bridging (%). (**I**) BMD (g/cm³). (**J**) BV/TV (%). (**K**) Tb.Th (mm). Results of quantitative µCT data analysis (ANOVA, followed by Tukey's post hoc test). Bars show mean ± SEM; N = 5–6.

The online version of this article includes the following source data for figure 2:

**Source data 1.** Source data for *Figure 2C–F and H–K*.

1 procollagen (P1NP). P1NP is considered a sensitive marker of bone formation (*Rosen, 2020*), and ELISA was performed using the serum samples collected at 14, 23, and 31 d post-femoral fracture. In WT animals, there is no difference in P1NP levels between the metformin-treated and PBS-treated mice at all three time points (*Figure 4B*). On the contrary, we observed significantly higher P1NP levels at all three time points post-fracture in the MKR mice treated with metformin compared to those treated with PBS (*Figure 4B*). Collectively, the data indicate that metformin can promote bone formation only under hyperglycemic conditions.

## Metformin regulates proliferation and lineage commitment of the bone marrow stromal cells (BMSCs) in MKR mice

Considering that the multipotent mesenchymal stem/progenitor cells (BMSCs) in bone are critical in maintaining bone quality, function, and regeneration, we then tested whether metformin stimulated BMSCs proliferation in MKR mice. As expected, hyperglycemic condition in MKR mice impaired the proliferation of their BMSCs, as indicated by CFU-F staining when compared to WT controls (*Figure 5A*). Administration of metformin in MKR mice successfully salvaged the BMSC proliferation capability and brought it back to the equivalent level as observed in the WT group. Under all three seeding densities, compromised CFU-F colony formation observed in the MKR group can be rescued by metformin treatment as shown in *Figure 5B*. However, metformin did not affect the proliferation of BMSCs in the WT group when compared to the PBS vehicle. It is noteworthy that metformin was not administrated to the culture. The daily metformin treatment in MKR mice prior to cell isolation appeared to be sufficient to protect the proliferation potential of BMSCs from the detrimental effects of hyperglycemia. BMSC lineage differentiation potential was also tested by ALP staining and von Kossa staining. After 14 d under osteogenic (*Danaei et al., 2011*) differentiation, in contrast to the weak ALP activity observed in the MKR-PBS group, while the MKR-met group showed compatible ALP activity to the WT groups (*Figure 5C and D*). ALP played a critical role in calcium crystallization and mineralization during bone formation; therefore, we speculated this aberrant ALP activity of the MKR-PBS group would further lead to an impaired bone mineralization. As expected, bone mineralization was barely detected in the MKR-PBS group after 21-day osteoblast differentiation visualized by von Kossa staining and metformin significantly enhanced bone mineralization in MKR mice when compared to the PBS-treated MKR animals (*Figure 5E and F*). Notably, the BMSCs from metformin-treated T2D mice could maintain the improved bone formation feature when re-exposed to the same hyperglycemic levels as in the T2D mice. BMSCs isolated from metformin-treated MKR mice and PBS controls were implanted in the recipient MKR mice (*Figure 5G*). Masson's Trichrome staining on the ossicles formed by the BMSCs from metformin-treated MKR mice showed greater bone formation than that of PBS-treated MKR mice (*Figure 5H and I*). The result indicated that the previous in vivo treatment of metformin protected the BMSCs from the hyperglycemic and inflammatory conditions in the MKR mice, thus they have better potential and capacity to form ossicles in vivo even in MKR recipient mice. Metformin could effectively rescue the impaired differentiation potential of BMSCs in MKR mice.

## Improved chondrogenesis of BMSC from metformin-treated MKR mice

Chondrogenesis and endochondral ossification are critical steps during the healing process after a bone injury. In order to examine metformin's effect on chondrogenesis during bone healing, we compared the cartilage deposition within fracture healing sites throughout the healing process (14 d, 23 d, and 31 d post-fracture). The cartilage deposition was significantly lower in the PBS-treated MKR mice compared to the PBS-treated WT mice at 14 d post-fracture (*Figure 6A and B*), which is consistent with the observation of SOX9 and PGC1α induction in the callus on days 12 and 21, respectively,

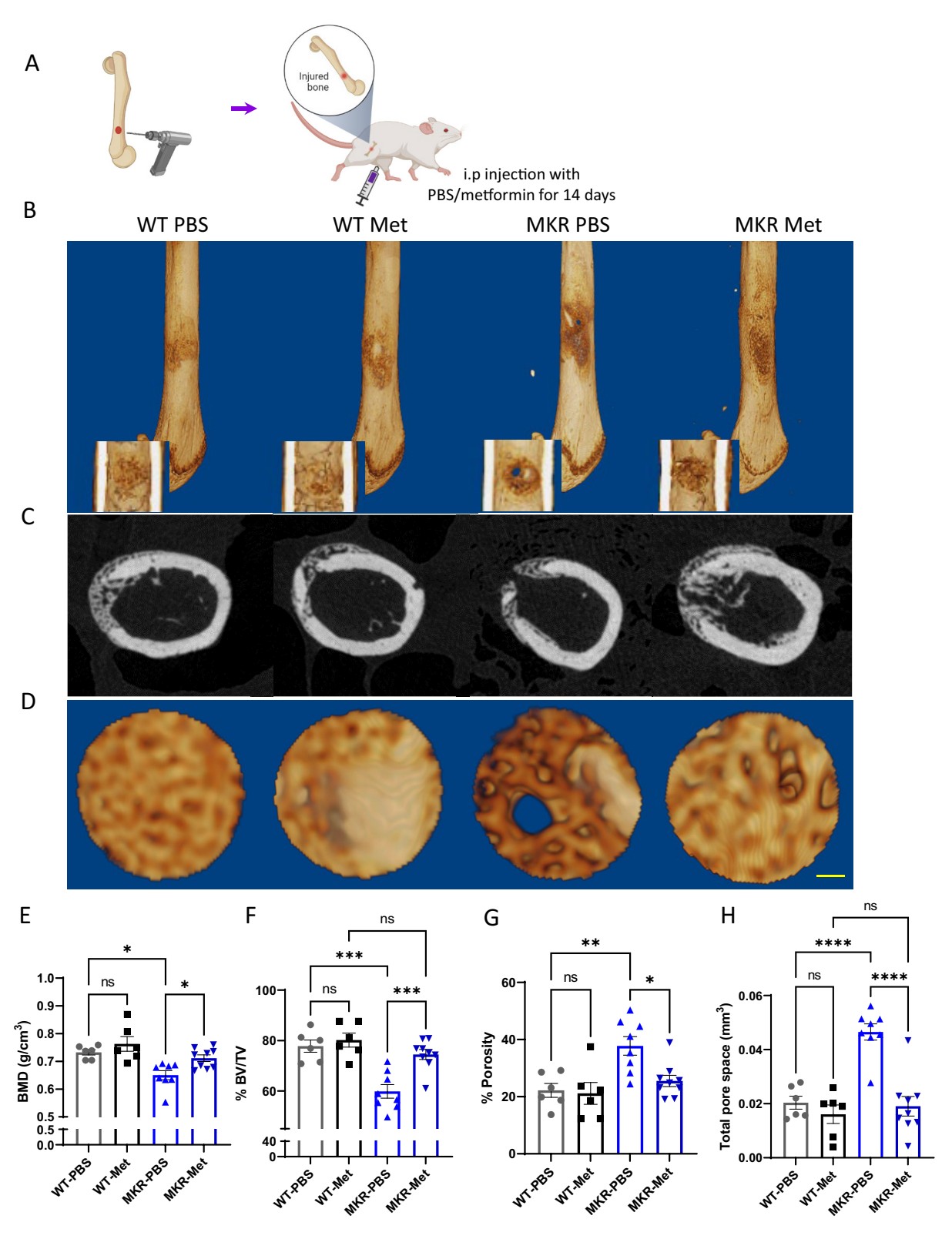

**Figure 3.** Improved healing in drill-hole bone repair model. (**A**) Schematic representation of the experimental design (created with BioRender.com). (**B**) Representative 3D images of mouse femurs with both exterior and interior general view at 14 d post-surgery. (**C**) Cross-plane images at the center of the drill site. (**D**) Representative 3D images within the drill site (scale bar, 100 μm). and (**E, H**) Results of quantitative μCT data analysis (ANOVA, followed

*Figure 3 continued on next page*

*Figure 3 continued*

by Tukey's post hoc test; bars show mean ± SEM; n = 6–9, *p<0.05, **p<0.01, ***p<0.005, **** p<0.0001). (**E**) Bone mineral density (BMD; g/cm³).
(**F**) Bone volume/tissue volume ratio (BV/TV; %). (**G**) Porosity (%). (**H**) Total pore space (mm³).

The online version of this article includes the following source data for figure 3:

**Source data 1.** Source data for *Figure 3E–H*.

of the metformin-treated MKR mice (*Figure 1—figure supplement 2*). As expected, no difference between the PBS- or metformin-treated animals was observed in the WT groups. On the other hand, metformin significantly promoted cartilage formation in MKR mice on day 14 post-fracture, and the trend continued till day 23 (*Figure 6A and B*). By day 31 post-fracture, except in the PBS-treated MKR mice, no discernible callus remained in any other groups (*Figure 6A*). To further investigate whether metformin modulates chondrogenesis of BMSCs. BMSCs were isolated for chondrogenesis culture in vitro. Only BMSCs isolated from the PBS-treated MKR mice failed to form the chondrocyte disc as shown in *Figure 6C* after 3-day chondrogenic culture. BMSCs isolated from the metformin-treated MKR mice could form the chondrocyte disc as well as the WT controls (*Figure 6C*).

## Discussion

Being the most commonly prescribed diabetic medication in the world, metformin's effects on bone healing in T2D patients remain unclear. To assess the impacts of metformin on fracture healing under hyperglycemic condition, we applied several classic bone fracture models in the T2D mice. Fracture healing is complex and involves the following stages: hematoma formation, granulation tissue formation, bony callus formation, and bone remodeling. During the bone healing, chondrogenesis lays down

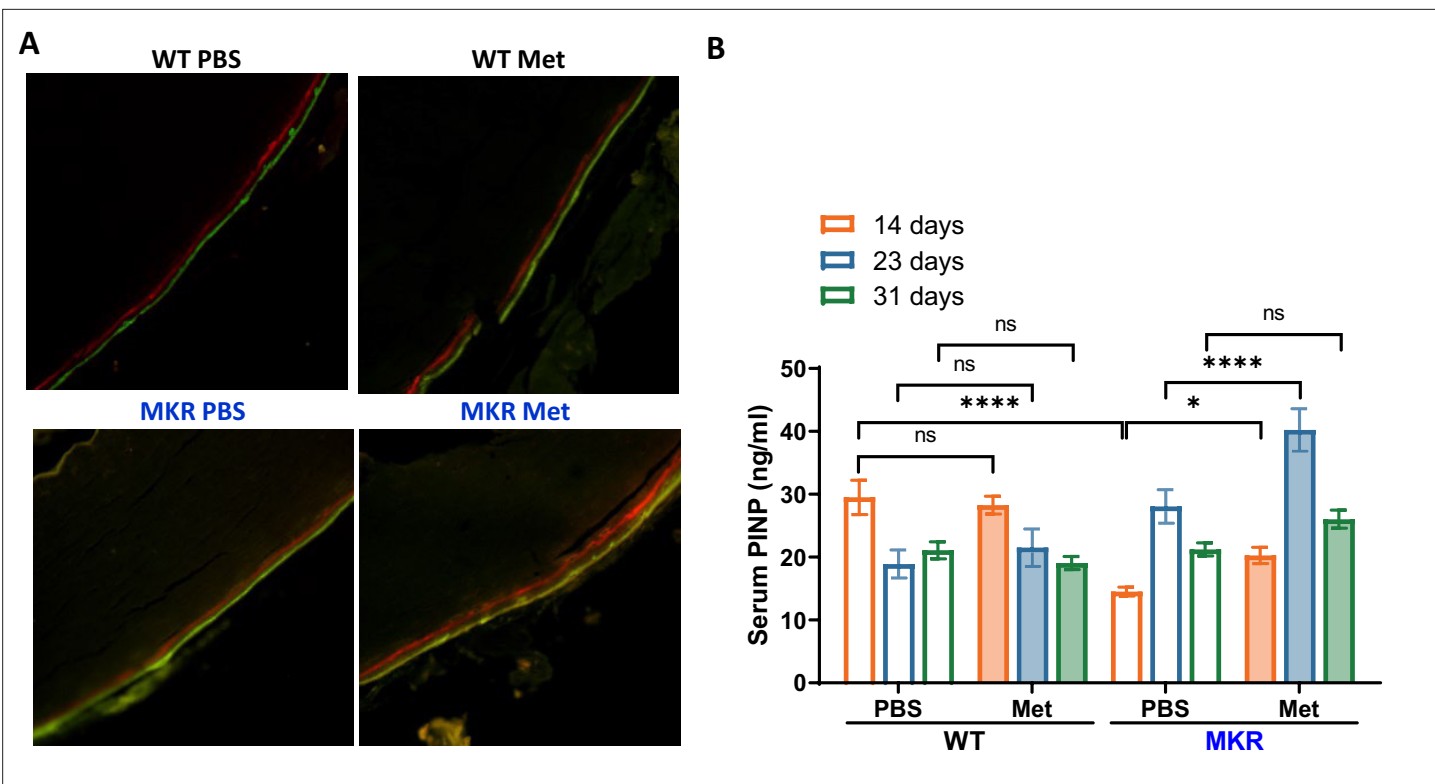

**Figure 4.** Metformin promotes bone formation. (**A**) Representative images of calcein double labeling in cortical periosteum of the femur mid-shaft. (**B**) Serum P1NP level (ng/ml) from each treatment group at 14, 23, and 31 d post- femoral fracture (ANOVA, followed by Sidak's post hoc test). Bars show mean ± SEM; N = 5–8, *p<0.05, ****p<0.0001.

The online version of this article includes the following source data for figure 4:

**Source data 1.** Source data for *Figure 4B*.

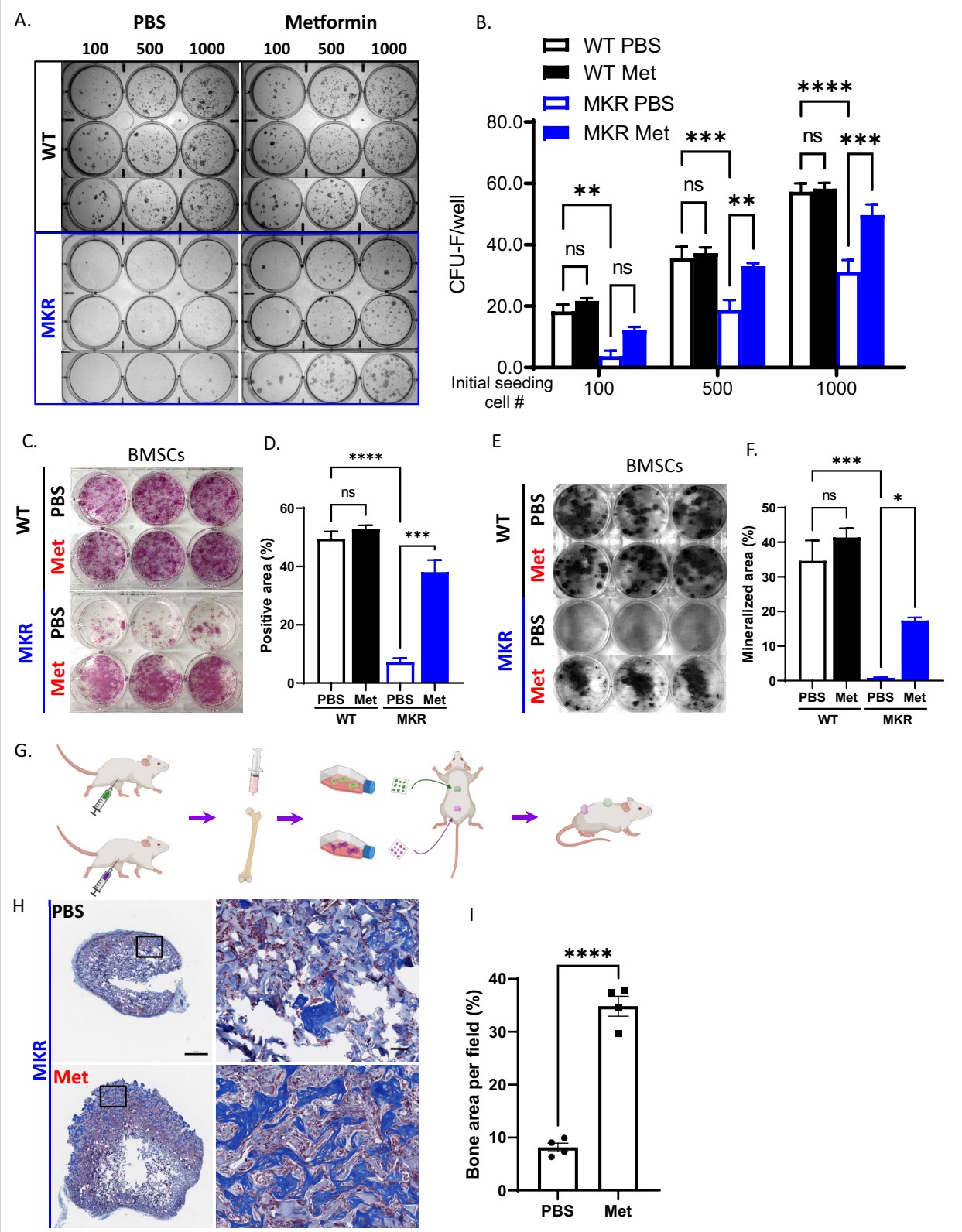

**Figure 5.** Improved osteogenesis of bone marrow stromal cell (BMSC) from metformin-treated MKR mice. (**A**) Primary BMSCs were isolated from animals that were treated with PBS or metformin in vivo for 14 d and were seeded at indicated density for CFU-F culture. (**B**) Results from quantitative analysis of colony counts measured using ImageJ. (**C**) Primary BMSCs from mice received 14-day treatment of PBS or metformin were plated in 6-well plates and cultured using differentiation medium and were tested for ALP activity, (**D**) Total ALP-positive area per well was measured using ImageJ.

*Figure 5 continued on next page*

*Figure 5 continued*

(**E**) von Kossa staining to examine mineralization. (**F**) Calculation of mineralized area. (**G**) Schematic representation of the experimental design. (**H**) Masson's Trichrome staining on the ossicle sections. (**I**) Percentage of bone area (blue) per field was measured using ImageJ. Bars show mean ± SEM. n = 6–9, *p<0.05, **p<0.01, ***p<0.005, ****p<0.0001.

The online version of this article includes the following source data for figure 5:

**Source data 1.** Source for *Figure 5B, D, F and I*.

a collagen-rich fibrocartilaginous network spanning the fracture ends and fibroblasts differentiate into osteoblasts so that cartilaginous callus eventually undergoes endochondral ossification (*Einhorn and Gerstenfeld, 2015*). Our results demonstrated that in all injury models tested metformin successfully rescued the delayed bone healing and remolding in the T2D mice by facilitating bone formations. Further cell culture studies demonstrated the mechanism of metformin's action at the cellular level by promoting the proliferation, differentiation, and lineage commitment of primary BMSCs. Taken together, metformin showed its potential as an effective drug for increasing the rate and success of bone healing in diabetic patients who are not taking metformin on a regular basis.

In all the bone fracture models studied in this study, metformin significantly enhanced bone-healing parameters in MKR mice. However, in the WT animals, quantitative analysis of images showed no difference between the PBS and the metformin treatment in terms of bone healing. These data suggest that metformin is only beneficial for bone healing under hyperglycemic conditions but does not enhance bone healing in WT animals without diabetes. In all BMSC-based essays, metformin was not administered to culture media in vitro but administered to animals in vivo before the BMSCs were isolated. Administration of metformin in MKR mice successfully salvaged the BMSC proliferation capability and lineage commitment and brought it back to the equivalent level as observed in the WT group. Interestingly, metformin did not affect the proliferation and lineage commitment of BMSCs in the WT group when compared to the PBS vehicle. These data are consistent with those obtained from bone fracture models, suggesting that metformin may exert its effects by normalizing hyperglycemia, glucose tolerance, and other metabolic disturbance under diabetic conditions and does not enhance bone healing in WT animals. In ovariectomized rats, impaired bone density and quality were significantly improved by the treatment of metformin (*Gao et al., 2010*). Taken together, it seems that metformin does not promote further bone growth under physiological conditions but helps to maintain bone homeostasis under pathological conditions such as hyperglycemia and lack of estrogen.

To our knowledge, there is very scarce amount of research on the direct effects of metformin on BMSCs. In an in vitro study using mouse bone marrow-derived multipotent mesenchymal stromal cells, metformin causes inhibition of proliferation and abnormalities of their morphology and ultrastructure and decreased IGF-2 secretion in the supernatant of the culture media (*Śmieszek et al., 2015*). In another paper, metformin added to culture is shown to promote the osteogenesis of BMSCs isolated from T2D patients and osseointegration when administered in rats via the AMPK/BMP/Smad signaling pathway (*Sun et al., 2022*). This discrepancy might be due to the different levels of metformin used in the culture system and different species of the experimental model. In one study (*Śmieszek et al., 2015*), inhibition of mouse BMSC proliferation was observed with 1, 5, and 10 mM of metformin. However, in another study (*Sun et al., 2022*), 125 µM was the optimal concentration to stimulate proliferation of BMSC in humans and rats. At concentrations >200 µM, metformin showed an inhibitory effect on BMSCs isolated from T2DM patients. Another possibility is that metformin does not work directly on BMSCs but requires other tissues, cells, and the in vivo context to exert its effect. Whether metformin has direct beneficial effects on BMSCs remains to be investigated.

As the most prescribed antidiabetic medicine and a drug poses benefits beyond the treatment of diabetes, the molecular and cellular mechanism of metformin demands a lot of research attention. The AMP-activated protein kinase plays a major role in its mechanism of action (*Rena et al., 2017*; *Foretz et al., 2014*); however, the exact signaling cascade and the direct molecular target of metformin remain unknown. Recently, low-dose metformin has been shown to target the lysosomal AMPK pathway through PEN2, a subunit of gamma-secretase (*Ma et al., 2022*). Clinically relevant concentrations of metformin bind PEN2 to activate AMPK pathways following glucose starvation. Loss-of-function analysis of PEN2 blunts AMPK activation, abolishes metformin-mediated reduction of hepatic fat content, and metformin's glucose-lowering effects. These novel findings indicate that metformin binds PEN2 directly and initiates a signaling route that intersects the lysosomal glucose-sensing

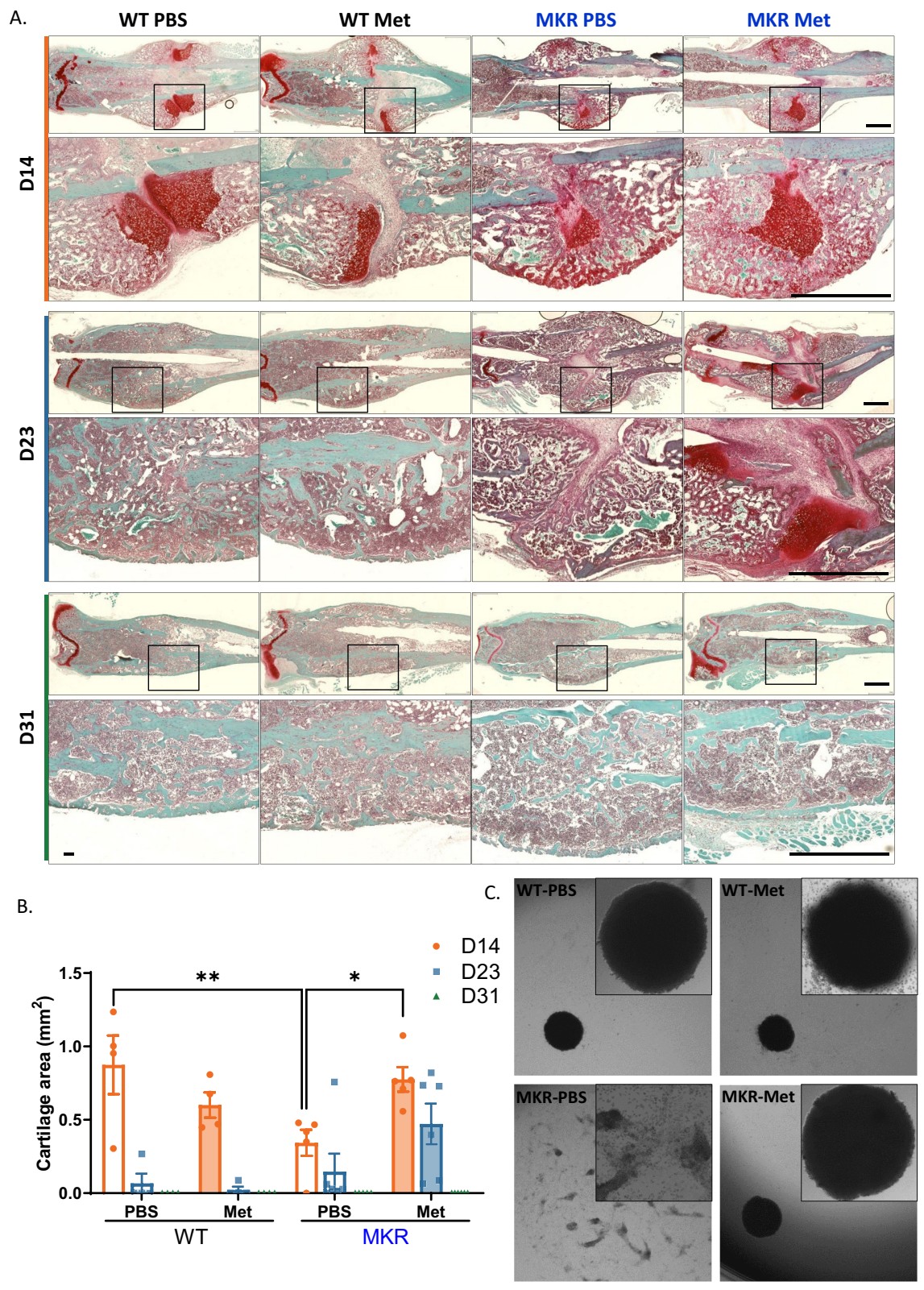

**Figure 6.** Improved chondrogenesis of bone marrow stromal cell (BMSC) from the metformin-treated MKR mice. (**A**) Safranin O staining of longitudinal femur sections from femoral fractures at different time points (scale bar, 1 mm). (**B**) Cartilage area at fracture site was measured using ImageJ. (**C**) Chondrocyte pellet culture of BMSCs from the PBS- or metformin-treated animals. Micromass culture were generated by seeding 5 μl of primary

*Figure 6 continued on next page*

*Figure 6 continued*

BMSCs ($1.6 \times 10^7$ cells/ml) in the center of 48-well plate and cultured under chondrogenic condition for 3 d (ANOVA, followed by Sidak's post hoc test). Bars show mean ± SEM. n = 4–6, *p<0.05, **p<0.01.

The online version of this article includes the following source data for figure 6:

**Source data 1.** Source data for *Figure 6B*.

pathway for AMPK activation. To our knowledge, there is no research done on the role of PEN2 in the beneficial effects of metformin on bone homeostasis and healing. It is very much worth the effort to investigate whether metformin binds PEN2 and utilizes the same signaling pathways in BMSCs. Another important aspect of the mechanism of metformin is its effects on osteoclasts. Metformin has been shown to inhibit osteoclast differentiation in various studies. Metformin inhibits osteoclastogenesis in ovariectomized mice by downregulating autophagy via E2F1-dependent BECN1 and BCL2 downregulation (*Xie et al., 2022*). In the osteoarthritis mouse model, metformin inhibits osteoclast activation and may exert its effects through the AMPK/NF-kappaB/ERK signaling pathway (*Guo et al., 2021*). In a rat osteonecrosis model, metformin inhibits osteoclast activity in the necrotic femoral head (*Park et al., 2020*). In an in vitro model of periodontitis, osteoclast formation as well as activity was inhibited by metformin in M-CSF and RANKL-stimulated monocyte cultures, probably by reduction in RANK expression (*Tao et al., 2021*). In another study, metformin inhibits osteoclast differentiation at a low concentration and promotes osteoclast apoptosis at a higher concentration (*Bian et al., 2021*). The effects of metformin on osteoclasts in various bone fracture models warrant future research.

As reviewed by *Roszer, 2011*, diabetes is accompanied by increased level of pro-inflammatory factors, reactive oxygen species (ROS) generation, and accumulation of advanced glycation end products (AGEs). The increased inflammatory state could result in apoptosis of osteoblasts and prolonged survival of osteoclasts, which lead to early destruction of callus tissue and impair bone fracture healing in diabetic patients. Thus, antagonizing inflammatory signal pathways and inhibition of inflammation may deserve greater attention in the management of diabetic fracture healing. There is substantial evidence supporting that metformin not only improves chronic inflammation by attenuating hyperglycemia but also has a direct anti-inflammatory effect. Targeting inflammatory pathways seems to be an important part of the comprehensive mechanisms of action of this drug (*Bharath and Nikolajczyk, 2021*). In addition to AMPK activation and inhibition of mTOR pathways, metformin acts on mitochondrial function and cellular homeostasis processes such as autophagy (*Bharath and Nikolajczyk, 2021*). Both dysregulated mitochondria and failure of the autophagy pathways affect cellular health drastically and can trigger the onset of metabolic and age-associated inflammation and diseases. For example, T-helper type 17 (Th17) cells, an important pro-inflammatory CD4+ T-cell subset secreting interleukin 17 (IL-17), has been suggested to play an essential role in the development of diabetes mellitus (*Abdel-Moneim et al., 2018*). Metformin can ameliorate the pro-inflammatory profile of Th17 by increasing autophagy and improving mitochondrial bioenergetics (*Bharath et al., 2020*). In addition, at day 21 post-fracture, the PGC1α in callus tissues isolated from the fracture site was significantly upregulated in the metformin-treated MKR mice. As reviewed by Halling and Pilegaard (*Halling and Pilegaard, 2020*), PGC-1α not only regulates mitochondrial biogenesis but also its function. PGC-1α-mediated regulation of mitochondrial quality may contribute to many age-related dysfunctions including insulin sensitivity. Anti-inflammation and enhancement of mitochondrial function could be very important means that metformin utilizes to facilitate bone formation and healing under hyperglycemic conditions.

Diabetic hyperglycemia has been suggested to play a role in osteoarthritis. The metabolic alterations in body fluid such as hyperglycemia could negatively affect the cartilage through direct effects on chondrocytes by stimulating the production of (AGE accumulation in the synovium *Li et al., 2022*). PPARγ is highly expressed in adipocytes, and the downregulation of PPARγ expression in the callus of metformin-treated MKR mice reflected the shift of mesenchymal cells' fate. In the T2DM mouse model, differentiation of growth plate chondrocytes is delayed and this delay may result from premature apoptosis of the growth plate chondrocytes (*Wongdee and Charoenphandhu, 2015*). Besides its effects on bone formation, there is also interest in studying the effects of metformin on chondrocytes, especially in the context of osteoarthritis development. Limited references show that metformin is protective against the development of osteoarthritis by reducing chondrocyte apoptosis and alleviating chondrocyte degeneration (*Yan et al., 2022b*; *Yan et al., 2022a*; *Feng et al., 2020*). Consistent

with the above reports, our data suggest that metformin promoted the cartilage formation in the endochondral ossification at day 14 post-fracture in the T2D mice. Moreover, metformin rescued the chondrocyte disc formation in BMSCs isolated from the T2D mice when compared to the PBS-treated control. Metformin also upregulated chondrocyte transcript factor SOX9 and in callus tissue isolated at the fracture site in the metformin-treated MKR mice. Sox-9 plays an essential role in the regulation of cartilage matrix production and cartilage repair (*Kolettas et al., 2001*). In addition, PGC1α was significantly upregulated in the metformin-treated MKR mice when compared to the PBS-treated MKR animals. PGC1alpha is important to maintain chondrocyte metabolic flexibility and tissue homeostasis. The loss of PGC1α in chondrocytes during OA pathogenesis resulted in the activation of mitophagy and stimulated cartilage degradation and apoptotic death of chondrocytes (*Kim et al., 2021*). The activation of PGC1α is a potential strategy to delay or prevent the development of OA. The cellular signaling pathways through which metformin exerts its protective effects in chondrocytes warrant further research.

In conclusion, our study demonstrated that metformin can facilitate bone healing, bone formation, and chondrogenesis in the T2D mice. The molecular mechanism of metformin's action demands further research in the hope to identify the specific therapeutic target to facilitate bone healing and repair in diabetic patients.

## Materials and methods

### Animals
All animal experiments were carried out with the compliance of the New York University Institutional Animal Care and Use Committee (IACUC). Hyperglycemic mouse model MKR (available from Jackson Laboratory; strain # 016618) breeders were generously provided by LeRoith and colleagues (*Fernández et al., 2001*). Friend Virus B (FVB) background wildtype (WT) breeders were ordered from Jackson Laboratory (Bar Harbor, ME).

### Bone injury models
#### Femoral closed fracture model
The gravity-induced Bonnarens and Einhorn fracture model was adapted here as previously described in order to establish a standard closed fracture (*Gerstenfeld et al., 2003*; *Bonnarens and Einhorn, 1984*). Briefly, 12-week-old male WT and MKR mice were anesthetized with a ketamine/xylazine cocktail, and a 1 cm sagittal incision was made at the right knee beneath the patella. A 3/8-inch-length 27-gauge needle was inserted into bone canal right between medial and lateral condyles. The needle end was cut, and the blunt end was pushed forward and buried between medial and lateral condyles to avoid tissue damage afterward. With the fixative needle inside of the femur canal, the animal was moved to the fracture apparatus. The right femur was placed over the two supports, and the blunt guillotine blade was dropped from a pretested height onto the femur to create sufficient force to cause the fracture. The wound was then closed with suture, and mice were randomly assigned to vehicle (PBS) or metformin (Met, 200 mg/kg body weight) daily treatment groups. In this study, metformin was dissolved in PBS for administration.

#### Radius non-union fracture model
12-week-old male WT and MKR mice were anesthetized using a ketamine/xylazine cocktail. A 0.5 cm coronal incision was made over the right radius. The brachioradialis and pronator teres were carefully separated with blunt surgical instruments to reveal the radius. A super sharp Stevens Tenotomy Scissor was used to cut at the middle of the radius and create the non-union radial fracture. The wound was then closed, and mice were randomly assigned to PBS or metformin daily treatment groups.

#### Femoral drill hole model
12-week-old male WT and MKR mice were anesthetized using a ketamine/xylazine cocktail. A 1 cm coronal incision was made over the right lateral femur. Quadriceps were carefully separated with blunt surgical instruments to reveal the femur. A drill bit (#66) was used to create a 0.8-mm-diameter hole in the femur. After closing the wound, mice were randomly assigned to PBS or metformin daily treatment groups.

## Tissue collection and processing

The micro computed tomography (μCT) and bone histomorphometry were utilized to assess static and dynamic indices of bone structure and formation. Briefly, bone injuries were introduced in animals as described above, and PBS and metformin treatment was administrated daily for the indicated time. After sacrificing the animal, injured bone samples were fixed in 10% buffered formalin for 48 hr, then rinsed with PBS before being analyzed by μCT. Bones were evaluated using a SkyScan 1172 high-resolution scanner (Brucker, Billerica, MA) with 60 kV voltage and 167 μA current at a 9.7 μm resolution and reconstructed using NRecon (V.1.6.10.2). Whole-scanned region was included as the volume of interest (VOI) (*Voigt et al., 2016*) from the femoral fracture model to generate a general 3D view of the femur fracture site. In the radius fracture model, a total of 3 mm (311 transverse anatomic slides) radius including the entire injury site was selected as VOI. In order to analyze only the fracture callus bone parameters, two sets of regions of interest (ROIs) (*Fernández et al., 2001*) were manually drawn within each VOI slides. The first set traced the external fracture callus (inclusion ROI), and the second set traced the cortical bones within the callus (exclusion ROI). In the femur drill hole model, a round-shaped ROI with 0.611 mm diameter (63 pixels) was surrounded in the center of injury hole throughout the depth where the callus was exhibit to generate the VOI consisting of callus within the drill site. VOIs from each animals were analyzed using CTan (V.1.13.2.1) to calculate the following morphometric parameters: bone mineral density (BMD), relative bone volume (BV/TV), trabecular thickness (Tb.Th), trabecular separation (Tb. Sp), porosity, and total pore space. CTVox (V.3.3.0.0) was used to generate the three spatial images of the VOI.

Callus bridging score was ranked from 0 to 3 (0 = no bridging; 1 = <33% bridging; 2 = >34% but <66% bridging; 3 = >67% till complete bridging). Callus bridging ratio was calculated by each animal's callus bridging score/3.

For double-labeling experiment, 10-week-old male WT and MKR mice were randomly assigned to receive PBS or metformin daily injections for 14 d. Mice also received intraperitoneal injections of calcein (10 mg/kg) on the 5th day and Alizarin Red (15 mg/kg) on the 12th day during the 14-day period. For histomorphometry and peripheral quantitative computed tomography, femurs were preserved in 70% ethanol until they were processed for plastic embedding in methyl methacrylate resin or decalcified for paraffin embedding.

Blood was collected by cardiac puncture after euthanasia and left at room temperature for 30 min before centrifuging at 200 × *g* for 10 min to separate sera.

## Cell culture and analysis

As previously described, after 14-day treatment of PBS or metformin in vivo, WT and MKR mice were sacrificed and proceeded with bone marrow primary cell culture. Excessive tissue was removed from the femur and tibia from each mouse and quickly rinsed with 70% ethanol and then followed by triple-cold PBS wash to ensure sterility. Bone marrow cells were flushed out using cold PBS and broken down into individual cell suspension by passing through a 19-gauge needle five times. Cells were then cultured in MEM Alpha Modification (α-MEM) medium supplemented with 10% fetal bovine serum (R&D Systems, USA), 100 μg/ml streptomycin, and 100 units/ml penicillin (Gibco, Grand Island, NY) in a 37°C, 5% (v/v) $CO_2$, humidified incubator. After removing non-adherent cells, the adherent cells were cultured as BMSCs for 7 d and sub-cultured for the following assays.

### Colony-forming unit fibroblastic assay (CFU-F)

Isolated BMSCs were seeded at 100, 500, and 1000 cells/well in 6-well plate, followed by 10-day culture with α-MEM complete medium. Cells were fixed with 10% buffered formalin, washed three times with PBS followed by staining with 0.25% wt/vol crystal violet solution. Plate images were captured using ChemiDoc XRS System (Bio-Rad Laboratories, Inc, Hercules, CA) and analyzed using ImageJ.

### Osteoblast differentiation

Isolated BMSCs were seeded at $6.4 \times 10^4$ cells/well into 6-well plates and cultured with osteogenic medium containing α-MEM complete medium supplement with 50 μg/ml l-ascorbic acid-2-phosphate and 10 mM β-glycerophosphate for 2–3 wk followed by ALP and von Kossa staining. Briefly, after 2 wk of osteogenic differentiation, cells were fixed and checked for alkaline phosphatase activity using

ALP kit (86R-1KT, Sigma) following the manufacturer's protocol. After 3 wk under osteogenic culture, cells were fixed with 95% ethanol and rehydrated through gradient ethanol to water. Cells were then carefully washed with water after being incubated with 5% silver nitrate solution at 37°C for 1 hr and exposed under UV light for 10 min. All plates' images were captured using ChemiDoc XRS System, and the ALP-positive area and mineralized regions were measured using ImageJ software.

## Chondrocytes differentiation and disc formation

Isolated BMSCs were further cultured and passaged twice using standard growth media of DMEM +10% FBS in order to enrich the cell number. A cell solution of $1.6 \times 10^7$ cells/ml was generated using StemPro Chondrogenesis Differentiation Kit (Thermo Fisher), and 5 µl droplets were applied in the center of 48-well plate wells for micro-mass culture. 3 hr later, warmed chondrogenic medium was overlayed over the micro-mass and the formation of osteogenic pellets was observed after 3 d of culture.

### In vivo ossicle formation assay

Adapted from a previous report (*Pettway and McCauley, 2008*) and as described above, BMSCs from different treatment groups were obtained and cultured using standard growth media of DMEM +10% FBS with customized glucose level. We measured and calculated the mean value of recipient MKR mice blood glucose level (N = 4, average glucose level = 436 mg/dl) and prepared the ex vivo culture medium accordingly to avoid glucose level change from ex vivo culture to the body fluid of the recipient MKR mice. Briefly, BMSCs from the PBS or metformin-treated mice were growing to 90% confluent, then $1.0 \times 10^8$ cells/ml cells solution was obtained, and 20 µl of this cell solution was soaked in a 4 mm × 4 mm gel foam and grafted at the flank of the recipient MKR mice subcutaneously. 4 wk after the implantation, the gel foams were dissected and processed for histological assays.

### Histology and image analysis

Femurs from fracture models and bone ossicles were decalcified using 10% EDTA for 2 wk. EDTA solution was refreshed every other day for the best decalcification efficacy. Tissues were then processed through an automatic tissue processor, followed by paraffin embedding. 5-µm-thick sections were cut from the paraffin-embedded tissues and subjected to H&E, Safranin O, and Mason's Trichrome staining, respectively. The total callus area and cartilage area within it were traced and analyzed using ImageJ software. The bone area ratio within the ossicles was also analyzed using ImageJ.

### Glucose test and glucose tolerance test

12-week-old WT and MKR mice were treated with PBS or metformin through daily intraperitoneal injection as previously described. Blood glucose level was recorded at the indicated time points for each mouse. After 14-day PBS or metformin injection, animals were fasted overnight for 16 hr by transferring into a clean new cage supplied with water. 20% (W/V) glucose solution was applied to the animals through oral gavage (1.6 g of glucose/kg body weight). Blood glucose was measured at the indicated time points till 2 hr post-glucose oral gavage.

### Quantitative PCR

A femoral closed fracture was performed as previously described. Callus tissue at the fracture site was collected and lysate in Trizol Reagent. RNA was extracted and purified using the RNeasy Plus kit. cDNA was prepared using a Reverse Transcriptase Kit, and PCR was then performed with a Bio-Rad CFX384 Real-Time PCR detection System. Primer information is given below:

> mβ-actin Fr: AGCCATGTACGTAGCCATCC;
> mβ-actin Rv: CTCTCAGCTGTGGTGGTGAA;
> mPGC1α Fr: GAATCAAGCCACTACAGACACCG;
> mPGC1α Rv: CATCCCTCTTGAGCCTTTCGTG;
> mSP7 Fr: GGCTTTTCTGCGGCAAGAGGTT;
> mSP7 Rv: CGCTGATGTTTGCTCAAGTGGTC;
> mRUNX2 Fr: CCTGAACTCTGCACCAAGTCCT;
> mRUNX2 Rv: TCATCTGGCTCAGATAGGAGGG;

mPPARγ Fr: GTACTGTCGGTTTCAGAAGTGCC;
mPPARγ Rv: ATCTCCGCCAACAGCTTCTCCT;
mC/EBPα Fr: CAAAGCCAAGAAGTCGGTGGACAA;
mC/EBPα Rv: TCATTGTGACTGGTCAACTCCAGC;
mSOX9 Fr: AAGAAAGACCACCCCGATTACA;
mSOX9 Rv: CAGCGCCTTGAAGATAGCATT

## Statistics

We used ANOVA when the study subjects included more than two groups, followed by post hoc tests as indicated in the legends. We used the two-tailed Student's $t$-test to compare the differences between the two experimental groups. A value of $p<0.05$ was considered statistically significant. Bars in the figures represent the mean ± SEM unless stated otherwise.

## Acknowledgements

This study was funded by the New York University Start-up and Career Enhancement Award to XL. The authors thank the support from the microCT core facility at the New York University College of Dentistry and Dr. Satoko Matsumura for her effort in the CFU-F assays.

## Additional information

### Funding

| Funder | Grant reference number | Author |
| --- | --- | --- |
| New York University | Start-up and Career Enhancement Award | Xin Li |

The funders had no role in study design, data collection and interpretation, or the decision to submit the work for publication.

### Author contributions

Yuqi Guo, Data curation, Formal analysis, Validation, Investigation, Methodology; Jianlu Wei, Methodology; Chuanju Liu, Methodology, Writing - review and editing; Xin Li, Conceptualization, Resources, Supervision, Funding acquisition, Investigation, Project administration, Writing - review and editing; Wenbo Yan, Conceptualization, Formal analysis, Methodology, Writing - original draft, Writing - review and editing

### Author ORCIDs

Chuanju Liu http://orcid.org/0000-0002-7181-8032
Xin Li http://orcid.org/0000-0002-7414-5734

### Ethics

This study was performed in strict accordance with the recommendations in the Guide for the Care and Use of Laboratory Animals of the National Institutes of Health. All of the animals were handled according to approved institutional animal care and use committee (IACUC) protocols (#141009 and IA16-00211) of the New York University. All surgery was performed under anesthesia, and every effort was made to minimize suffering.

### Decision letter and Author response

Decision letter https://doi.org/10.7554/eLife.88310.sa1
Author response https://doi.org/10.7554/eLife.88310.sa2

## Additional files

### Supplementary files
• MDAR checklist

## Data availability

All data generated or analysed during this study are included in the manuscript and supporting file.

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
