## [Editor Report]

This fundamental work advances our understanding of the effects of metformin on bone healing in hyperglycemic conditions. The evidence supporting the conclusion is convincing, using three different types of bone fracture models in type-2 diabetes (T2D) mice. This paper is of potential interest to skeletal biologists, orthopaedic surgeons, and endocrinologists who study the effects of metformin on fracture healing.

---

## [Decision Letter]

**Decision letter after peer review:**

Thank you for submitting your article "Metformin regulates bone marrow stromal cells to accelerate bone healing in diabetic mice" for consideration by *eLife*. Your article has been reviewed by 3 peer reviewers, and the evaluation has been overseen by a Reviewing Editor and Mone Zaidi as the Senior Editor.

Essential revisions:

This is a generally well-executed set of study, the data from which are appropriate and largely supports the conclusions. The manuscript would be strengthened by (1) including more discussion/explanation on the mechanisms underlying metformin's effects on healing; and (2) improving the writing of the manuscript for clarity and accuracy.

*Reviewer #1 (Recommendations for the authors):*

The following are some specific comments:

1. Abstract can be improved in writing to clearly demonstrate the positive effects of Metformin in bone healing. What is the conclusion from bone ossicle implantation experiments? The conclusion of the study is also not clear in the abstract.

2. The direct and indirect effects of Metformin in T2D mice can be better explained in the discussion. The conflicting literature also should be cited and explained better.

3. Why bone density can be improved by Metformin in ovariectomized rats? Do ovariectomized rats also have hyperglycemia or increased inflammation? This needs to be better explained in the discussion.

4. The quantification data in Figure 1D are not well described in the figure legend. It is not clear whether they are histomorphometric analyses or MicroCT analyses?

5. Figure 1D needs to be better labeled and explained. Statistical analyses also need to be clearly stated.

6. Figure 2G appears to show delayed union of the bone in T2D mice. Is the ratio of bony union in T2D mice improved by Metformin treatment? What is the n in the experiment shown in Figure 2?

7. Does Type 2 diabetes also affect osteoclasts? What are the effects of Metformin on osteoclasts in healing?

8. The results from Figure 5G and H appear to show that MSCs derived from Met treated mice differentiate better and form more bone in diabetic mice. Why? This needs to be better explained since these diabetic mice have hyperglycemia and increased inflammation in vivo. Are similar numbers of cells implanted into the mice in this experiment?

9. Figure 6, what type of fracture is examined here? Are these the same data sets as those shown in Figure 1 or 2?

10. Persistent cartilage is not good for healing. It is not clear whether there is any residue cartilage left at day 31. This should be illustrated in the quantification data in Figure 6B to show that the rate of the healing is no different in Met treated mice but the size of the callus is larger.

11. Figure 6A, bone and cartilage tissues are not very well separated in the histologic staining. Are there any immature bone tissues in the diabetic mice?

12. Why *Sox9* and PGC1-α expressions are presented in the supplemental data?

13. Is it possible to perform biomechanical testing on these samples to determine the functionality of the bone following metformin treatment?

*Reviewer #2 (Recommendations for the authors):*

The paper's conclusions and results are strong, but more attention needs to be paid to the introduction and description of the prior literature and understanding of the potential mechanism and signaling pathway of metformin's function in the MKR mouse model bone healing. In 2022, a Nature article entitled: "Low-dose metformin targets the lysosomal AMPK pathway through PEN2" was published. It would be good if authors can discuss this paper in the Discussion section to elaborate on the potential specific targets and signaling pathway of metformin in bone healing.

*Reviewer #3 (Recommendations for the authors):*

The authors, in their research manuscript titled 'Metformin regulates bone marrow stromal cells to accelerate bone healing in diabetic mice', dissected the role of Metformin in bone healing under type-2 diabetics conditions. The authors used three classic bone fracture models to assess the impacts of Metformin in bone healing under hyperglycemic conditions. In all three models, Metformin treatment showed bone formation. At the cellular level, the authors showed the effect of Metformin on promoting bone healing using BMSCs in vitro. The authors in the paper demonstrated that Metformin promotes bone growth only in hyperglycemic conditions. The experiments were appropriately well-defined and carried out to support the role of Metformin in bone healing. The use of three different bone-defective rat models to study the role of Metformin in skeletal tissues is convincing. The authors may address the following comments to further strengthen their paper.

1. The abstract mentioned that Metformin promotes bone healing, bone formation, and chondrogenesis, but in the title, it was mentioned as bone healing. Please clarify in the paper.

2. Why was an intraperitoneal injection of Metformin performed in the animal models? The caudal artery injection in the rats' tails would also be the preferred model for bioactive molecule delivery.

3. What was the half-life of Metformin under in vivo conditions?

4. Please include whether Metformin was dissolved in PBS or any other solvent in the paper.

5. The authors may include in the discussion about the effect of Metformin on decreasing or inhibiting osteoclast activity if it does.

6. What is the selective mechanism activated by Metformin under hyperglycemic conditions for bone formation? Please include the proposed or possible mechanisms for this effect if it is unavailable.

---

## [Author Response]

Essential revisions:Reviewer #1 (Recommendations for the authors):The following are some specific comments:1. Abstract can be improved in writing to clearly demonstrate the positive effects of Metformin in bone healing. What is the conclusion from bone ossicle implantation experiments? The conclusion of the study is also not clear in the abstract.

Revision has been made in the abstract to clarify the results.

2. The direct and indirect effects of Metformin in T2D mice can be better explained in the discussion. The conflicting literature also should be cited and explained better.

More information is added to the discussion to better explain this.

3. Why bone density can be improved by Metformin in ovariectomized rats? Do ovariectomized rats also have hyperglycemia or increased inflammation? This needs to be better explained in the discussion.

The lack of estrogen due to ovariectomy contribute to bone loss. Despite that ovariectomized rats do not have hyperglycemia, metformin is likely to improve the bone density of OVX rats through targeting osteoclasts and bone resorption. More information is added into the discussion to explain this.

4. The quantification data in Figure 1D are not well described in the figure legend. It is not clear whether they are histomorphometric analyses or MicroCT analyses?

It is clarified in the legend now.

5. Figure 1D needs to be better labeled and explained. Statistical analyses also need to be clearly stated.

We revised the entire Fig1D and the time points and treatment groups are clear. Statistical is also added into the legend.

6. Figure 2G appears to show delayed union of the bone in T2D mice. Is the ratio of bony union in T2D mice improved by Metformin treatment? What is the n in the experiment shown in Figure 2?

The administration of metformin significantly improved the ratio of bony union in mice with type 2 diabetes (T2D) at 23 days post-fracture, as depicted in Figure 2H. Additionally, both bone mineral density (BMD) and bone volume fraction (BV/TV) indicated improved bone quality at the callus site for both time points. The sample size for this injury model was 5 to 6 mice, and this information has been included in the figure legend.

7. Does Type 2 diabetes also affect osteoclasts? What are the effects of Metformin on osteoclasts in healing?

Type 2 diabetes influence on osteoclast activity remain controversial and obscure in patients but several studies have reported that metformin suppressed osteoclast activity in mice. We also have showed that metformin inhibited the elevated osteoclastogenesis and activity in MKR mice. A section of the possible effects of metformin on osteoclast has been added to the discussion.

8. The results from Figure 5G and H appear to show that MSCs derived from Met treated mice differentiate better and form more bone in diabetic mice. Why?

The two groups in comparison are met-treated MKR mice and PBS-treated MKR mice. The result indicated that BMSCs from the MKR mice primed with metformin for two weeks are protected from the hyperglycemic and inflammatory conditions, thus they have better potential and capacity to form ossicles in vivo even in MKR recipient mice.

This needs to be better explained since these diabetic mice have hyperglycemia and increased inflammation in vivo.

More explanation of the results have been added to the result section.

Are similar numbers of cells implanted into the mice in this experiment?

Yes, similar numbers of cells were implanted into MKR mice in this experiment.

9. Figure 6, what type of fracture is examined here? Are these the same data sets as those shown in Figure 1 or 2?

We examined the cartilage formation in femoral fracture model. It’s the same set of animals we evaluated the callus area in Figure 1.

10. Persistent cartilage is not good for healing. It is not clear whether there is any residue cartilage left at day 31. This should be illustrated in the quantification data in Figure 6B to show that the rate of the healing is no different in Met treated mice but the size of the callus is larger.

We agree. There is no cartilage left on day 31, that’s why we only included the earlier time points (14 days and 23 days) in Figure 6B. We’ve update Figure 6B to include day 31 cartilage area quantifications.

11. Figure 6A, bone and cartilage tissues are not very well separated in the histologic staining. Are there any immature bone tissues in the diabetic mice?

It is possible that immature bone tissues exist in the healing site in the diabetic mice.

12. Why Sox9 and PGC1-α expressions are presented in the supplemental data?

We think they are belonging to the Supplemental data of the transcription factors related to adipogenesis, osteogenesis or chondrogenesis.

13. Is it possible to perform biomechanical testing on these samples to determine the functionality of the bone following metformin treatment?

This is a very good suggestion. We will consider conducting this test with new mice in the future studies but the samples used in the current study can’t be used for biomechanical testing anymore.

Reviewer #2 (Recommendations for the authors):The paper's conclusions and results are strong, but more attention needs to be paid to the introduction and description of the prior literature and understanding of the potential mechanism and signaling pathway of metformin's function in the MKR mouse model bone healing. In 2022, a Nature article entitled: "Low-dose metformin targets the lysosomal AMPK pathway through PEN2" was published. It would be good if authors can discuss this paper in the Discussion section to elaborate on the potential specific targets and signaling pathway of metformin in bone healing.

A section of the possible mechanisms of metformin including the above-mentioned article has been added to the discussion and included here:

“Recently, low-dose metformin has been shown to target the lysosomal AMPK pathway through PEN2, a subunit of γ-secretase [19]. Clinically relevant concentrations of metformin binds PEN2 to activate AMPK pathways following glucose starvation. Loss of function analysis of PEN2 blunts AMPK activation, abolishes metformin-mediated reduction of hepatic fat content, and metformin’s glucose-lowering effects. These novel findings indicate that metformin binds PEN2 directly and initiates a signaling route that intersects the lysosomal glucose-sensing pathway for AMPK activation. To our knowledge, there is no research done on the role of PEN2 in the beneficial effects of metformin on bone homeostasis and healing. It is very much worth the efforts to investigate whether metformin binds PEN2 and utilizes the same signaling pathways in BMSCs.”

Reviewer #3 (Recommendations for the authors):The authors, in their research manuscript titled 'Metformin regulates bone marrow stromal cells to accelerate bone healing in diabetic mice', dissected the role of Metformin in bone healing under type-2 diabetics conditions. The authors used three classic bone fracture models to assess the impacts of Metformin in bone healing under hyperglycemic conditions. In all three models, Metformin treatment showed bone formation. At the cellular level, the authors showed the effect of Metformin on promoting bone healing using BMSCs in vitro. The authors in the paper demonstrated that Metformin promotes bone growth only in hyperglycemic conditions. The experiments were appropriately well-defined and carried out to support the role of Metformin in bone healing. The use of three different bone-defective rat models to study the role of Metformin in skeletal tissues is convincing. The authors may address the following comments to further strengthen their paper.1. The abstract mentioned that Metformin promotes bone healing, bone formation, and chondrogenesis, but in the title, it was mentioned as bone healing. Please clarify in the paper.

Bone healing involves multiple steps/processes, which includes bone formation and chondrogenesis. The abstract has been edited to reflect this and information on the bone healing has been added to the first paragraph of the discussion.

2. Why was an intraperitoneal injection of Metformin performed in the animal models? The caudal artery injection in the rats' tails would also be the preferred model for bioactive molecule delivery.

Metformin is primarily administered orally in tablet for human patients. We selected IP injection because the taste of metformin tends to reduce the daily water consumption in mice, the variation of water consumption could also lead to inaccurate dosing of metformin. Caudal artery injection in rats’ tail is a preferred model for bioactive molecule delivery, however when working with mice, the size of their tails is technically challenging and requires unnecessary restrain of mice for a longer time.

3. What was the half-life of Metformin under in vivo conditions?

Metformin is not metabolized and is excreted unchanged in the urine, with a half-life of ~5 h.

4. Please include whether Metformin was dissolved in PBS or any other solvent in the paper.

Metformin was dissolved in PBS and we’ve added this in the method part.

5. The authors may include in the discussion about the effect of Metformin on decreasing or inhibiting osteoclast activity if it does.

A section of the possible effects of metformin on osteoclast has been added to the discussion and included here:

“Another important aspect of the mechanism of metformin is its effects on osteoclasts. Metformin has been shown to inhibit osteoclast differentiation in various studies. Metformin inhibits osteoclastogenesis in ovariectomized mice by downregulating autophagy via E2F1-dependent BECN1 and BCL2 downregulation [20]. In osteoarthritis mouse model, metformin inhibits osteoclast activation and it may exert its effects through AMPK/NF-kappaB/ERK signaling pathway [21]. In a rat osteonecrosis model, metformin inhibits osteoclast activity in the necrotic femoral head. [22]. In an in vitro model of periodontitis, osteoclast formation as well as activity was inhibited by metformin in M-CSF and RANKL stimulated monocyte cultures, probably by reduction of RANK expression [23]. In another study, metformin inhibits osteoclast differentiation at a low concentration and promotes osteoclast apoptosis at a higher concentration [24].”

6. What is the selective mechanism activated by Metformin under hyperglycemic conditions for bone formation? Please include the proposed or possible mechanisms for this effect if it is unavailable.

The possible mechanisms of metformin including direct regulation of PEN2 in osteoblasts have been added to the discussion.